# Inhibition of *Escherichia* Virus MS2, Surrogate of SARS-CoV-2, via Essential Oils-Loaded Electrospun Fibrous Mats: Increasing the Multifunctionality of Antivirus Protection Masks

**DOI:** 10.3390/pharmaceutics14020303

**Published:** 2022-01-27

**Authors:** Joana M. Domingues, Marta O. Teixeira, Marta A. Teixeira, David Freitas, Samira F. da Silva, Shafagh D. Tohidi, Rui D. V. Fernandes, Jorge Padrão, Andrea Zille, Carla Silva, Joana C. Antunes, Helena P. Felgueiras

**Affiliations:** 1Centre for Textile Science and Technology (2C2T), Campus de Azurém, University of Minho, 4800-058 Guimaraes, Portugal; joana.domingues@2c2t.uminho.pt (J.M.D.); pg35037@alunos.uminho.pt (M.O.T.); martaalbertinateixeira@gmail.com (M.A.T.); samifs@gmail.com (S.F.d.S.); ruidvfernandes@gmail.com (R.D.V.F.); padraoj@2c2t.uminho.pt (J.P.); azille@det.uminho.pt (A.Z.); joana.antunes@2c2t.uminho.pt (J.C.A.); 2Centre of Biological Engineering (CEB), Campus de Gualtar, University of Minho, 4710-057 Braga, Portugal; davidsfreitas@ceb.uminho.pt (D.F.); carla.silva@ceb.uminho.pt (C.S.); 3Digital Transformation Colab (DTx), Campus de Azurém, University of Minho, 4800-058 Guimaraes, Portugal; shafagh.tohidi@dtx-colab.pt

**Keywords:** electrospun fibers, antiviral essential oils, physisorption versus blending, SARS-CoV-2 surrogate, viral inhibitors

## Abstract

One of the most important measures implemented to reduce SARS-CoV-2 transmission has been the use of face masks. Yet, most mask options available in the market display a passive action against the virus, not actively compromising its viability. Here, we propose to overcome this limitation by incorporating antiviral essential oils (EOs) within polycaprolactone (PCL) electrospun fibrous mats to be used as intermediate layers in individual protection masks. Twenty EOs selected based on their antimicrobial nature were examined for the first time against the *Escherichia coli* MS2 virus (potential surrogate of SARS-CoV-2). The most effective were the lemongrass (LGO), Niaouli (NO) and eucalyptus (ELO) with a virucidal concentration (VC) of 356.0, 365.2 and 586.0 mg/mL, respectively. PCL was processed via electrospinning, generating uniform, beadless fibrous mats. EOs loading was accomplished via two ways: (1) physisorption on pre-existing mats (PCLaEOs), and (2) EOs blending with the polymer solution prior to fiber electrospinning (PCLbEOs). In both cases, 10% *v*/*v* VC was used as loading concentration, so the mats’ stickiness and overwhelming smell could be prevented. The EOs presence and release from the mats were confirmed by UV-visible spectroscopy (≈5257–631 µg) and gas chromatography-mass spectrometry evaluations (average of ≈14.3% EOs release over 4 h), respectively. PCLbEOs mats were considered the more mechanically and thermally resilient, with LGO promoting the strongest bonds with PCL (PCLbLGO). On the other hand, PCLaNO and PCLaELO were deemed the least cohesive combinations. Mats modified with the EOs were all identified as superhydrophobic, capable of preventing droplet penetration. Air and water-vapor permeabilities were affected by the mats’ porosity (PCL < PCLaEOs < PCLbEOs), exhibiting a similar tendency of increasing with the increase of porosity. Antimicrobial testing revealed the mats’ ability to retain the virus (preventing infiltration) and to inhibit its action (log reduction averaging 1). The most effective combination against the MS2 viral particles was the PCLbLGO. These mats’ scent was also regarded as the most pleasant during sensory evaluation. Overall, data demonstrated the potential of these EOs-loaded PCL fibrous mats to work as COVID-19 active barriers for individual protection masks.

## 1. Introduction

At the end of 2019, a novel strain of coronavirus, severe acute respiratory syndrome coronavirus 2 (SARS-CoV-2), was identified in China. Compared to previous outbreaks, COVID-19 was deemed extremely dangerous due to its high rate of contagion, severe symptoms and elevated mortality rates [1,2]. According to the World Health Organization (WHO), more than 259 million confirmed cases of COVID-19 and more than 5 million deaths have been registered worldwide, to this date [3]. Although the pandemic has a global dimension, the virus impact on each human being is different, and the severity of the disease (mild, moderate, severe or fatal) varies depending on the age and pre-existent pathologies (e.g., cardiovascular, respiratory, cerebrovascular and immunodeficiency) of each individual [4]. The most common symptoms are fever, cough, fatigue, sputum production, headache, hemoptysis, diarrhea, dyspnea and lymphopenia. On average, these symptoms start manifesting after five days of incubation [2,4]. The airborne transmission of the SARS-CoV-2 virus between people occurs through contact (direct or indirect) with air containing infected aerosols or respiratory droplets [5]. Worldwide, several preventive measures have been proposed, including social distancing, disinfection of objects, self-hygiene (e.g., hand washing) and the use of personnel protective equipment (e.g., face masks and respirators) [6]. In several countries, the use of protective masks is mandatory within closed public spaces. However, most of this equipment has a passive action against COVID-19 [5]. There have been reports stating that the virus can remain infectious in standard surgical masks for up to seven days [7]. The permanence of the active virus in masks thus reinforces the need for more innovative and effective protective options to help contain its spreading [8].

Essential oils (EOs) are a complex mixture of volatile and aromatic chemical compounds (e.g., terpenes, phenols, alcohols, aldehydes, ethers and ketones) extracted from different parts of plants (e.g., buds, flowers, stems, leaves, seeds, twigs, roots, fruits, bark and wood flowers, seeds, leaves, fruits and roots) [9,10,11,12]. These have the function of defending the host from microbial invasion, manifesting antibacterial, antifungal and insecticide activities, aside from antiviral [13,14,15]. Furthermore, EOs have anti-inflammatory, immunomodulatory and bronchodilatory properties [16]. They are widely available in nature, have low water solubility, strong lipophilicity and low toxicity [9,10,14]. Their composition, and consequently inherent properties, are dependent on the harvest date [17], cultivation conditions [18] and geographical area [19], plant variety [20], plant or part of the plant age [21], extraction system [22], among others. EOs have shown important effects against several pathogenic viruses by penetrating the viral membranes and triggering their rupture [16]. Therefore, the use of EOs may be an advantageous approach when dealing with the SARS-CoV-2 [16,23,24]. Garlic oil (GLO), for example, can inhibit the activity of the angiotensin-2 converting enzyme protein, leading SARS-CoV-2 to lose its host receptor and attack its main protease, the PDB6LU7 protein, hindering virus maturation, proliferation and colonization [25]. It is also known that inhaling certain EOs together with steam has a positive impact on bronchiolitis, colds, rhinosinusitis, allergies, flu and asthma [23]. Several EOs have been studied and applied in biomedicine [9,26], including lemongrass (LGO), Niaouli (NO) and eucalyptus (ELO) [11]. LGO has citral (3,7-dimethyl-2,6-octadienal) as the most abundant and biologically active compound within its composition [27]. This oil has antioxidant characteristics, anti-inflammatory and analgesic properties, and good activity against a plethora of fungal, bacterial and viral species [27,28,29]. Furthermore, this oil is frequently used to treat fever, flu, colds and pneumonia [29,30]. NO is extracted from *Melaleuca quinquenervia* and is rich in 1,8-cineole (eucalyptol). NO is described as having a strong antimicrobial action and to intervene in the treatment of respiratory infections and by attenuating coughs and colds [31,32]. Like NO, ELO has good antimicrobial properties, specifically antiviral [33], and is frequently used in traditional folk medicine to treat respiratory tract disorders and infections [34,35]. It is also composed essentially of 1,8-cineole, limonene, α-pinene, γ-terpinene, and α-terpineol [33]. To this date, neither of these EOs has been examined for its potential against the SARS-CoV-2 virus, either alone or as part of human personnel protective equipment.

Despite the antiviral potential of EOs, their sensitivity to external factors (temperature, light and oxygen) and their volatile nature tend to limit their application [10]. Nevertheless, recent studies have described the protective effect of polymeric fibers on the EOs, as a way to ensure their stability and to preserve their chemical composition [11,14]. Electrospinning is one of the most common techniques employed in the production of fine polymeric fibers (ranging the nanometer scale), with great flexibility, excellent mechanical properties and a continuous three-dimensional, intricated network [36,37,38,39]. This technique is simple, inexpensive, effective, and versatile. More importantly, films resulting from electrospinning are porous and possess a highly interconnected architecture that allows easy incorporation of biomolecules of interest [38], such as EOs [40].

In this study, we propose the production of polycaprolactone (PCL) fibrous mats loaded with selected antiviral EOs, via electrospinning, for prospective applications in individual protective masks (as intermediate layers). PCL is a synthetic biodegradable polymer produced by the ring-opening polymerization of ε-caprolactone monomer using a wide range of catalysts [36]. Due to its biocompatibility, excellent mechanical strength, miscibility with other polymer solutions, slow degradation rate and non-toxicity, PCL is widely used in biomedicine and tissue engineering systems [11,36]. Indeed, there are several medical devices (i.e., ‘Monocryl’ sutures, ‘Capronor’ birth control device, etc.) made of this material that have been approved by the Food and Drug Administration (FDA) for human uses [41]. PCL networks have also shown abilities to serve as delivery platforms for a variety of biomolecules, namely EOs [11,40]. Therefore, the main objective of the present study is to explore the antiviral potential of selected EOs, while integrated into PCL electrospun mats, against the *Escherichia coli* MS2 virus (a potential surrogate of SARS-CoV-2). This bacteriophage is a non-pathogenic virus, consisting of an icosahedral capsid and used to study human pathogenic viruses such as Influenza A and B, as well as SARS-CoV [42,43]. Such parallelism is possible due to the similarity in resistance to antimicrobial agents and the ease of preparation and testing between them [42]. Although there are some studies that explore the potential of polymeric PCL networks loaded with EOs for other purposes [38,44], to the authors’ knowledge, none has proposed such an endeavor yet.

## 2. Experimental Section

### 2.1. Materials

Twenty EOs were purchased from *Folha d’Água* (Santo Tirso, Portugal). Table 1 provides detailed information about each one. Their selection was made based on the EOs inherent antibacterial activity, reported previously by our team [11]. *E. coli* bacteriophage MS2 (ATCC 15597B1) and respective *E. coli* host (ATCC 15597) were supplied from American Type Culture Collection (ATCC). Both bacteriophage and host were incubated in ATCC^®^ Medium 271 (M271). A water-based solution composed of 10 g/L tryptone, 1 g/L yeast extract and 8 g/L sodium chloride was autoclaved at 121 °C, and then aseptically combined with an aqueous solution containing 1 g/L of glucose, 0.294 g/L of calcium chloride and 0.01 g/L of thiamine. Petri dishes containing M271 agar, at 5% *w*/*v* (top layer) and 15% *w*/*v* (bottom layer) concentration, were used as solid media. PCL (Mn 80,000), chloroform (CHL) and dimethyl formamide (DMF) were purchased from Sigma-Aldrich and used without further purification.

### 2.2. Minimum Bactericidal Concentrations (MBC) and Virucidal Concentrations (VC)

MBCs were determined using the broth microdilution method described by Wiegand et al. [45], which adapts the standard published by the Clinical and Laboratory Standards Institute (CLSI) and the European Committee on Antimicrobial Susceptibility Testing (EUCAST) [46].

EOs were diluted in M271 at 50–1.25% *v*/*v*, which corresponded to an average maximum concentration of 481.80 ± 71.10 mg/mL and an average minimum concentration of 12.05 ± 1.78 mg/mL; maximum and minimum concentrations for each oil were dependent on their inherent density (Table 1). Working solutions prepared at 50, 40, 30, 20, 10, 5, 2.5, 1.25% *v*/*v* of EOs in M271 (defined based on the averaged MICs obtained for the 20 selected oils against *E. coli* bacterium [11]) were added (50 μL) to each column of the 96-well plates. Then, to each of these wells, 50 μL of the *E. coli* (host) suspension prepared at 1 × 10^7^ colony forming units (CFUs)/mL in M271 were added. *E. coli* suspension without EOs and culture media were used as controls. Absorbance readings at a wavelength of 600 nm (EZ READ 2000 Microplate Reader, Biochrom Ltd., Cambridge, UK) were performed before and after plate incubation at 37 °C and 120 rpm for 24 h, for minimum inhibitory concentration (MIC) determinations. The bacterium was then cultured on M271 agar plates (preparation of aliquots of 10 μL of each bacteria suspension diluted from 10^1^ to 10^5^ in phosphate-buffered saline solution, PBS) at MIC and at a concentration in its vicinity (higher and lower) and the number of CFUs/mL were counted. Absence of viable colonies on agar after 24 h of culture at 37 °C established MBC rates.

The VC of each EO against the bacteriophage MS2, prepared at 1 × 10^7^ plaque-forming units (PFUs)/mL in M271, was determined using the same concentrations and volumes selected for MIC testing. After 24 h incubation at 37 °C and 120 rpm, all suspensions contacting with the bacteriophage were cultured on bacterium-seeded two-layer agar plates (diluted from 10^1^ to 10^5^ in M271). The agar plates were prepared by inoculating bacterium in the top layer (liquid state) of agar. The host is required for the virucidal effect to be detectable and PFUs/mL to be counted. Absence of bacteriophage plates on agar after 24 h of culture at 37 °C established VC rates. The three most effective EOs against the bacteriophage MS2 (factoring the effect on the host) were highlighted from the group and used in the subsequent experiments. Those same three EOs were also examined for their composition via solid-phase micro-extraction followed by gas chromatography-mass spectrometry (SPME-GC-MS), by exposing the SPME fiber (100 μm polydimethylsiloxane) to the vapor phase above the EOs sample (at 1 mg/mL) for a period of 4 h at 35 °C (temperature of exhaled air).

### 2.3. Electrospun Fibrous Mats Production and EOs Loading

PCL was solubilized at 14 wt.% in CHF/DMF (9/1 *v*/*v*) under constant stirring of 150 rpm and room temperature (RT), for 24 h (control). Viscosity was assessed using a viscometer Brookfield DV-II+Pro (from Hadamar-Steinbach, Germany, with spindle 21, processed at 15 rpm at 25.3 °C during 10 min, with 30 s time points), while conductivity was determined using a Thermo Scientific Benchtop Meter (Orion Versa Star Pro, Waltham, MA, USA). A fixed voltage of 23 kV was applied to the steel capillary needle with an inner diameter of 21 Gauge (G). The solution feeding rate was set at 0.7 mL/h, with an aluminum collecting sheet being positioned at 26 cm from the needle tip for fiber recovery. Temperature and relative humidity (RH) were controlled and maintained at 20–22 °C and 60–65%, respectively.

EOs were incorporated within the PCL mats via two ways: (1) blending prior to fiber extrusion (labeled as PCLbEOs) and (2) physical adsorption after obtaining the fibers (labeled as PCLaEOs). In strategy (1), the EOs at 10% VC were blended for 24 h with a freshly prepared PCL solution (150 rpm and RT). To prevent interferences with the EOs stability, solutions were maintained protected from light until extrusion. These were characterized (viscosity and conductivity) and processed via electrospinning following the same parameters established for PCL, the only exception being the voltage which was fixed at 26 kV. In approach (2), samples of 11 mm of diameter were cut using a metal puncher from PCL mats and immersed in EO’s alcoholic solutions prepared at 10% *v*/*v* VC for 24 h at 150 rpm (orbital shaking), protected from light. In the end, mats were collected, and unbound molecules were eliminated with a 5 min ethanol washing at 150 rpm (orbital shaking). Ethanol traces on the mats were eliminated after drying for 24 h in a desiccator with controlled RH of ≈41%.

### 2.4. Morphological Examinations

Micrographs of the PCL, PCLaEOs and PCLbEOs fibrous structures were acquired with an accelerating voltage of 10 kV using a field emission gun scanning electron microscope (FEG-SEM, NOVA 200 Nano SEM, FEI Company, Hillsboro, Oregon, USA). Mats were initially coated with a thin film (10 nm) of Au-Pd (80–20 wt%) using a 208HR high-resolution sputter coater (Cressington Company, Watford, UK) coupled to an MTM-20 Cressington High Resolution Thickness Controller. Average fiber diameters were determined by conducting 100 fiber measurements on three micrographs of each type, simulating the diameter distribution with a log-normal function [47]. Images at a magnitude of 10,000× were used and processed using the ImageJ software (version 1.51j8, National Institute of Health, Bethesda, MD, USA). The number of layers and porosity level of the mats were analyzed using the Python 3.8 software (Python, Amsterdam, The Netherlands).

### 2.5. Detection of EOs via Fourier-Transform Infrared Spectroscopy with Attenuated Total Reflectance (ATR-FTIR)

An IRAffinity-1S, SHIMADZU spectrophotometer (Kyoto, Japan), with an ATR accessory (diamond crystal), was used to investigate the PCL, PCLbEOs and PCLaEOs mats’ chemical groups. For each mat, a total of 200 scans were performed at a spectral resolution of 2 cm^−1^, over the wavenumber range of 700–4000 cm^−1^.

### 2.6. EOs Loading and Release Kinetics

EO loading was determined via two ways, using UV-visible spectroscopy: (1) indirectly, in adsorbed surfaces (PCLaEOs), by measuring the absorbance of the loading (*t* = 0 h and *t* = 24 h) and washing solutions, using a UV–1800 UV-vis spectrophotometer (Shimadzu, Kyoto, Japan) and subtracting that amount to *t* = 0 h, this way allowing to determine the mass of EOs integrated into each mat; and (2) directly, analyzing the oil-blended mats (PCLbEOs) with a UV-2600 UV-vis spectrophotometer (Shimadzu, Kyoto, Japan) and an integrating sphere (ISR-2600Plus) with a film holder for transmittance analysis (film rectangles with 6 × 2 cm^2^ sliced from the mats centered regions). In each strategy, calibration curves were made for the tested EOs to identify relevant peaks.

EOs release kinetics from the engineered mats was mapped over a period of 4 h (recommended maximum time for mask use). Samples of 11 mm diameter (including control without oils) were immersed in 1 mL of distilled water (dH_2_O) and incubated at 150 rpm and RT for 30 min, 1, 2 and 4 h. At the end of each period, the solution was collected and absorbance measurements (200–500 nm) were conducted using the UV-1800. The % of EOs released from the mats was determined based on absorbance differences between time 0 h and the defined testing periods. Any alterations in the absorbance of the control samples (PCL) were considered and subtracted from the final absorbance for each tested sample.

### 2.7. Quantification of EOs Release by GC-MS

The EOs (LGO, NO and ELO) release study was carried out via headspace (HS) by exposing the SPME fiber (100 μm polydimethylsiloxane) to the vapor phase above the sample matrix, followed by GC-MS evaluation. The SPME fiber was inserted in the middle of the vial containing the sample and exposed for 4 h at 35 °C (temperature of exhaled air) [48].

The samples were quantified by GC-MS using manual injection of the SPME fiber. Gas chromatographic analyses were carried out using a Varian 4000 system (Walnut Creek, CA, USA) with a split/splitless injector coupled to a mass spectrometer. Injections were operated at 250 °C in the split mode 1:10 using an Rxi-5Sil MS (Restek, Bellefonte, PA, USA) column (30 m × 0.25 mm, and 0.25 μm film thickness), with a column-head pressure of 7.3 psi using helium as carrier gas. The oven temperature started at 45 °C and was held for 5 min; the temperature increased until 250 °C at a rate of 7 °C/min. A full scan mode (50–750 *m*/*z*) was applied for the identification of the target compound. The mass spectrometer (MS) was operated in electron ionization (EI) mode at 70 eV with total ion chromatogram (TIC) detection mode for quantitative determination and S/N ratio of 5. Calibration curves of the oils were accomplished using the same testing conditions of the samples (temperature and time). Each time point was evaluated separately, and all the measurements were performed in triplicate. The amount of oil was determined by the integration of the peaks from chromatograms and quantified against the calibration curves.

### 2.8. Qualitative Assessment of EOs Release: Sensory Evaluation

Odor intensity assessments were conducted on PCL, PCLaEOs and PCLbEOS by a group of 45 people, a number considered acceptable for such a trial [49]. In this sensory evaluation, the general impression of a group of untrained volunteer participants, such as students and employees of the University of Minho, with no experience as panelists in sensory evaluation, was collected [50]. Before starting the sensory assessment, each participant received a questionnaire (Appendix A) to fill out during the sensory assessment and its purpose was explained. Each participant placed the samples 1 to 2 cm distant from the nose and breathed in the odor for 30 s. Intervals of 30 s were made between samples of sensory evaluation to reduce cross-adaptation and perception influence for decision making [51]. Descriptive, discriminatory (such as ordering test and pair comparison) and affective analyses were conducted [49]. For each odor, the subjective intensity perceived by each participant was analyzed. After inhaling each odor, the participants were asked to rank the mats in ascending order of intensity and to indicate the intensity of each mat using a scale from 0 to 5, where 0 = not perceptible, 1 = weakly/not perceptible, 2 = moderately perceptible, 3 = clearly perceptible, 4 = strongly perceptible, and 5 = very strongly perceptible [52]. Consecutively, a test of comparison by pairs between EOs loading strategies (blended or absorbed) was carried out, with the participants having to select the mat that exuded the greatest odor intensity from each pair. Finally, participants performed an affective analysis by classifying mats according to their unpleasant or pleasant odor quality using a hedonic scale from −4 to 4, where −4 = extremely unpleasant, −3 = moderate unpleasant, −2 = unpleasant, −1 = slightly unpleasant, 0 = not unpleasant but not pleasant, +1 = slightly pleasant, +2 = pleasant, +3 = moderate pleasant, and +4 = extremely pleasant [51,53].

### 2.9. Thermal Stability

Thermal gravimetric analyses (TGA) were conducted by weight loss monitoring with an increase of temperature in the range of 25–500 °C, at a heating rate of 10 °C/min under a dynamic nitrogen atmosphere and flow rate of 200 mL/min (inert environment) on an STA 7200 Hitachi^®^ (Fukuoka, Japan), using aluminum pans. The initial mass of each sample was established at 3.44 ± 0.32 mg. Results were plotted as % of weight loss vs temperature. Differential scanning calorimetry (DSC) evaluations were conducted in a DSC 822 Mettler Toledo (Columbus, OH, USA). Samples of 3.04 ± 0.17 mg were submitted to a temperature range of 0–500 °C heated at a rate of 10 °C/min, under a dynamic nitrogen atmosphere, and flow rate of 200 mL/min (inert environment). DSC curves were plotted as heat flow vs. temperature.

### 2.10. Mechanical Performance

The tensile strength and elongation at break of the unloaded and EOs-loaded PCL mats were evaluated using a Hounsfield H5KS dynamometer (Artilab, Kerkdriel, The Netherlands) associated with the QMAT Materials Testing & Analysis software, following the standard ASTM D5035. Three rectangular-shaped specimens of 6 cm long and 2 cm width were cut from each film. The average thickness of the samples was determined at 0.09 ± 0.01 mm using a handheld analogical micrometer with a dial indicator from Mitotoyo (ref. 2046F, Senhora da Hora, Portugal) with a resolution of 0.01 mm, 10 mm pressing area and 18 Pa of pressure. The gauge length and grip distance were established at 2 cm. The crosshead speed was 10 mm/min and the selected load cell was 2.5–250 N, used with a load range of 25 N and a pre-load of 0.2 N. Experiments were performed at RT.

### 2.11. Wettability and Degree of Swelling

Water contact angle measurements were conducted in a Contact Angle OCA 15, Data Physics apparatus (Filderstadt, Germany) connected to a video-based drop shape analyzer OCA15 plus software, following the standard ASTM-D7334–08. Droplets of 10 μL of dH_2_O were used to evaluate the mats’ wettability via the sessile drop measurement. Six measurements were performed per type of sample. Angles were recorded immediately after drop contact with the surface. The mats’ degree of swelling (DS) was determined by measuring the weight of the samples before and after immersion in dH_2_O for 4 h (recommended maximum time for mask use) at 150 rpm. DS was calculated using the following equation (Equation (1)) and reported in %:(1)$$\mathrm{DS}\;(\%)=\frac{\left(m_{w}{-m}_{d}\right)}{m_{w}}\times 100\;$$
where *m_w_* (mg) is the weight of the wet film, *m_d_* (mg) is the weight of the dry film.

### 2.12. Air and Water-Vapor Permeabilities

The mats’ ability in promoting air exchange was determined according to standard ISO 9237 using an FX 3300 LABOTESTER III, TEXTEST Instrument (Schwerzenbach, Switzerland). An air pressure of 40 Pa (differential pressure defined on EN ISO 14683:2019 as the minimum required for particle filtering of medical face masks) was applied onto six different and equidistant points on three samples of each type, under an area of 5 cm^2^. Air permeabilities were reported in L/cm^2^.

On water vapor permeability examination (standard BS 7209:1990), mats were placed on top of cylindrical cups containing 46 mL of dH_2_O for 4 h. The evaporation of water through the test sample was determined by weighing the cup before and after the testing period. Examinations were performed at RT of 19–21 °C and 61–64% RH (triplicates). Water vapor permeability standard testing fabric (defined in BS 7209:1990) was used as a control. The water vapor transmission rate (WVTR) and the water vapor permeability index (*I*) were determined using Equation (2):(2)$$WVTR=\frac{24\Delta W}{A\Delta t},\;and\;I=\frac{{WVP}_{s}}{{WVP}_{r}}\times 100\;$$
where ∆*W* is the difference in the water weight (g) before and after the 4 h test, *A* is the inner area of the cup (mm), ∆*t* is the exposure time (h), WVPs is the water vapor permeability of the samples and *WVPr* is the water vapor permeability of reference (standard testing fabric).

Commercial surgical masks with ≥95% (certified in conformity with EN149:2001+A1:2009, no brand, China) and of ≥98% protection (certified in conformity with EN1483:2019, Pharmia, Portugal) were used for comparison purposes.

### 2.13. Bacteriophage Permeability

An adaptation of the Kirby-Bauer method was implemented to examine the mats’ permeability or lack of it against the *Escherichia* virus MS2. Agar plates seeded with the host bacteria were prepared and left to solidify (see Section 2.2). After, samples of 1.1 cm in diameter were gently placed on top of the agar and loaded with droplets of 5 µL containing the bacteriophage prepared at 1 × 10^7^ PFUs/mL in M271. This volume was defined to mimic high-risk interactions between COVID-19 patients, for instance, in which droplet transmission after coughing or sneezing may occur. Plates were incubated for 4 and 24 h at 37 °C. Images were collected and examined for the presence (permeable) or absence (impermeable) of phage plaques.

### 2.14. Bacteriophage Contact Inactivation

Bacteriophage solutions were prepared at 1 × 10^7^ PFUs/mL in M271 and inoculated in the form of 50 µL loads (a volume defined to cover the entire surface without leaking to the exterior) on top of unloaded and EOs-loaded PCL mats of 1.1 cm in diameter. Samples were incubated for 4 h at RT (mimicking mask use conditions). In the end, mats were submerged in 5 mL of M271 and vortexed for 3 min for thorough washing of the surfaces and detachment of bacteriophage plaques. Finally, bacteriophage-containing suspensions (diluted from 10^1^ to 10^5^ in M271) were cultured on bacterium-seeded two-layer agar plates and incubated for 24 h at 37 °C. Grown plaques were counted and converted into log reduction (mean ± S.D.). For the purposes of eliminating the effect of the EOs on the host, equal experiments were conducted with *E. coli*. However, instead of washing and preparing diluted suspensions for plate culturing with M271, PBS was used.

## 3. Results and Discussion

### 3.1. MBCs and VCs of Selected Eos

Twenty EOs (Table 1) were tested for their MBCs and VCs against the bacterium *E. coli* (host) and the bacteriophage MS2, respectively (Table 2). From the tested group, only six EOs were found effective against the bacteriophage. As such, for the purposes of this work, MBCs of those oils were the only reported values (MICs and MBCs were equal).

Data established LGO, NO and OO as the three EOs most effective from the tested group. However, while for LGO and NO that represent a 40% *v*/*v* dilution from the “mother” solution, OO corresponds to a 50% *v*/*v* ratio, which implied a larger amount of OO oil for fighting the same concentration of bacteriophage. From the other active EOs, TTO, CO and ELO, only ELO required a 40% *v*/*v* ratio as well (denser oil from the group); both TTO and CO demanded a 50% *v*/*v* dilution. Considering the final application and the attention that should be given to prospective production costs, the smallest amount of oil is always a benefit. For that reason, OO was replaced by ELO, while TTO and CO were neglected as potential replacement options. Additionally, since ELO was not effective against the host *E. coli* strain, all observed impacts on the bacteriophage were not affected by host interference.

LGO is an antiviral EO, abundant in citral (≈63% of its composition, Appendix A), which action against viral pathogens is time-dependent: the longer the exposure, the more oil molecules bind to the viral particles, and the more effective becomes the antimicrobial action. It is not yet clear, but data supports the premise that LGO inhibits the infectious nature of the virus by directly binding to the virus capsid. Through this interaction, LGO generates an antimicrobial coating around the viral particles, preventing specific binding between virus and host cell receptors, thus successfully inhibiting infection spreading [54]. Aside from its obvious antiviral features, LGO is also frequently used to relieve fever, flu, colds and pneumonia-related symptoms [29,30]. Considering many of the symptoms that characterized these diseases have been reported on COVID-19-infected patients [2,4], the selection of LGO for further testing is more than justified. Both NO and ELO benefit from the inherent antimicrobial properties of their main component, the 1,8-cineole (≈45% for NO, Appendix A, and ≈54% for ELO, Appendix A) and its synergisms with less concentrated compounds, such as limonene, α-pinene and α-terpineol [33,55]. EOs containing a high percentage of 1,8-cineole are deemed less cytotoxic than those of the same family that are governed by other compounds [56]. Yang et al., by analyzing the effect of 1,8-cineole on bronchitis virus, demonstrated that the natural component was harmless to the host cells. More importantly, it was seen that even though its effect was moderate against the virus prior to contact with the host cells, it grew exponentially once the virus penetrated the host, by inactivating important RNA functions and preventing binding with host cell proteins [57]. On its own, 1,8-cineole is capable of blocking many of the specific binding domains of the viral particles. Yet, for an active role against virus prior to cell penetration, 1,8-cineole requires the assistance of the other compounds, like the monoterpene limonene, which has been reported to inactivate viruses in pre-treatments prior to host cell infection [58]. As LGO, NO and ELO have also been involved in the treatment of respiratory tract disorders and infections, attenuating coughs and colds [31,32,34,35].

The VCs for each of the selected EOs ranged between 356 and 586 mg/mL (Table 2). Compared to other published reports, these concentrations may be considered elevated. However, when working with natural extracts, it is almost impossible to establish relationships between investigations, since the antimicrobial activity of the natural compounds is intimately dependent on the extraction methods, culture conditions, site of production, etc. [9]. Here, our goal was to explore the EOs extracted from the Portuguese company *Folha d’Água*, which have never been tested against viral particles. In previous works by our team, we have seen that selected EOs from this company are highly effective against Gram-positive and Gram-negative bacteria while loaded onto fibrous constructs. Our goal is to continue exploring these natural-origin molecules’ antimicrobial profiles. Further, these increased VCs may also be explained by the difficulty in eliminating bacteriophages. It has been shown that bacteriophages are more difficult to eliminate than a model virus (i.e., influenza A) using EOs [59]. This is the first report on the use of EOs against the bacteriophage MS2 and, as such, no comparison can be made.

### 3.2. Electrospun Mats Morphology

PCL and PCL blended with LGO, NO and ELO were processed by electrospinning in the form of fibrous mats (detailed information on the blends’ properties, namely viscosity and conductivity, is provided in Table 3). In parallel, PCL mats were modified post-production with EOs via physical adsorption. In both loading strategies, the EOs were incorporated into the blends or adsorbed onto the mats at 10% VC. Concentrations superior to 10% VC were not considered since the obtained mats presented a very sticky, wet-like appearance and the scent from the oils was too intense. The authors were aware that this reduction could compromise the antiviral performance of the EOs; yet a balance between virus elimination and user comfort was required. Further, it has been shown that EOs at concentrations below their optimal against microbial pathogens can still perform above average, reporting log reductions of 1 and 2 after 4 h exposure [11].

FEG-SEM micrographs were acquired from PCL (control), PCLbEOs (blended EOs) and PCLaEOs (physically adsorbed EOs) revealing very smooth, uniform, and bead-free morphologies (Figure 1). The different EOs incorporation strategies did not alter these traits of the fibers’ morphology. The average fiber diameters of the electrospun mats ranged from 0.75 ± 0.18 (PCLbLGO) to 2.72 ± 0.93 µm (PCL). The incorporation of the EOs led to a decrease in fiber diameter as compared to neat PCL mats. Indeed, blending PCL with the selected EOs before electrospinning resulted in fibers ranging from 0.75 ± 0.18 (PCLbLGO) to 1.43 ± 0.40 µm (PCLbNO), whereas the average diameter of EOs adsorbed fibers varied between 1.46 ± 0.42 (PCLaLGO) to 1.83 ± 0.59 µm (PCLaNO). An increase of the fibers’ diameter would be expected by the incorporation of these natural-origin biomolecules [60]. However, the opposite was observed. By reducing the viscosity of the blends (only exception PCLbNO, Table 3), jet stretching may have been facilitated, thus generating smaller fiber diameters [61]. Between blends, PCLbNO promoted the largest fiber diameters by generating higher viscoelastic forces that resisted the axial stretching during whipping [62,63]. An inverse correlation between β-caryophyllene content and solution viscosity has been established [64]. From the group, NO had the lowest content on β-caryophyllene (elemental datasheet provided by the supplier), which could explain these observations. Further, by using ethanol as a solvent for the adsorbed Eos, shrinkage of the mats likely occurred, thus being accompanied by the reduction of the fibers’ diameters. In both cases (blending and adsorption), the use of biomolecules with a plasticizing-like effect induced a small but perceptive crimping of the fibers (particularly on NO and ELO modified mats, Figure 1) as the oriented polymer chain conformation, frequently obtained via electrospinning, relaxes to a more isotropic coil-like structure [65].

The morphology and average fiber diameter can be significantly affected by the solutions’ conductivity as well. PCL conductivity was determined at 0.225 µS/cm (Table 3). Polymer solutions’ conductivities are dependent on the solvent used and solute concentration. CHF belongs to the low electrical conductivity group [66]. Yet, DMF is known to alter the distribution of the fibers obtained on the collector, as it raises electrical conductivity and dielectric constant and lowers surface tension, giving rise to smaller fiber diameters [67]. Electrospinning is dependent on the Coulomb force between the charges on the surface of the fluid and the external electrical field [68]. Even though the addition of EOs decreased the conductivity of the polymeric blends (reduction of free ions movement due to new interactions being established), as expected [69], by raising the external electrical voltage from 23 kV (PCL) to 26 kV (PCLbEOs), the conductivity influence on the fibers’ diameters could be surpassed. Many have reported the reduction of nanofibers’ diameters with increasing applied voltages, as they raise the solutions electrical force and inherent stress, facilitating spinning [14,68].

The porosity and surface area of fibrous mats dictate their breathability abilities for prospective uses in face masks. In electrospinning, porosity can be increased by raising the applied voltage [70]. Our data corroborate that premise; the incorporation of EOs within the PCL blends resulted in an almost 16.2–21.5% increase in porosity compared to PCL. Further, by reducing the fiber diameters during EOs adsorption (ethanol influence), the spaces between fibers also enlarged, facilitating access to the inner regions of the mats and, this way, increasing the porosity.

### 3.3. Confirmation of EOs Presence via ATR-FTIR

Spectra of the unloaded and EO-loaded PCL mats were collected (Figure 2). Strong bands associated with the base polymer were quickly identified. Between 2950–2850 cm^−1^, the CH_2_ symmetric and asymmetric stretching vibrations of PCL were detected. At 1721 cm^−1^, a peak assigned to the carbonyl (C=O) stretching vibrations were observed. At 1293 cm^−1^ another peak was detected for the C–O and C–C stretching vibrations and was linked to the crystalline phase of the polymer, while those same vibrations associated with the amorphous region were recognized at 1162 cm^−1^. Finally, at 1239 cm^−1^, the C–O–C asymmetric stretching vibrations of PCL were identified [71].

Recognition of EOs loaded onto polymeric scaffolds via FTIR has been performed mostly through peaks at 1577 and 1458 cm^−1^, frequently assigned to the aromatic ring C=C skeleton vibration of an aromatic substance and to the C–OH absorption bending vibration from the alcohol moieties of the EOs, respectively [11,26]. Here, however, they were mostly imperceptible in the spectra. Still, the presence of EOs led to an increase in the PCL characteristic peaks height and area (Appendix A). Further, a very clear peak at 1471 cm^−1^ was detected on mats containing EOs. This may be associated with the -OCH_3_ asymmetric bending vibrations of the oils [72]. At 1683 cm^−1^, the peak related to the CH=CH vibrations of the citral group was also detected on LGO-containing mats [73]. Identification of EO-related peaks confirms the successful incorporation of these natural-origin biomolecules within the electrospun mats.

### 3.4. EOs Loading and Release

The successful loading of LGO, NO and ELO onto/into PCL fibrous mats during mat processing was evaluated by UV-visible spectroscopy, over wavelengths ranging from 200 to 500 nm, to include all relevant phenomena of light absorption/scattering from these samples. On the other hand, the quantification of the EOs volatile compounds release was mapped by GC-MS, with a triggering temperature of 35 °C (defined as the average temperature of exhaled air) [48], for a period of 4 h (Table 4).

Absorbance scans of increasing EO solution concentrations allowed the evaluation of EO presence and contribution to the obtained spectra, additionally enabling the determination of EO-specific calibration curves that were then used to calculate EO mass within the films (Appendix A). EO adsorption started with the immersion of pristine PCL mats into a solution containing EO at its loading concentration (10% VC, Appendix A) for 24 h. LGO-containing samples revealed substantially higher EO presence in the supernatant than NO-loaded films. LGO-adsorbed mats carried a peak of light absorbance characteristic of citral, the main active component of the EO, at a wavelength of 237 nm [74]. NO and ELO amount adsorbed to PCL nanofiber mats were similarly examined but considering the 1,8-cineole contribution to the overall spectra, the main constituent of both EOs [33,55]. However, after rinsing in ethanol, PCLaNO was the sample that release the highest amount of EO, followed by ELO and LGO. Samples were left to dry at RT in a desiccator, with PCLaLGO films retaining the higher EO amount at the end of sample processing. On another hand, by analyzing the films in their solid-state (Appendix A), the wavelength of 237 nm of the LGO-blended mats also carried a peak of light absorbance characteristic of citral [74]. Moreover, a shift of the ≈280 nm absorption band of the free oil to ≈300 nm of loaded PCLbLGO films reinforces the occurrence of a strong interaction of the oil with the PCL electrospun fibers, as well as a successful entrapment/distribution of phenolic molecules of the LGO within the polymeric matrix [26]. No distinctive peak could be perceived from the direct visualization of EO contribution of the spectra of NO- and ELO-blended films [33,55]. Still, the examination of peak height (using the data of unloaded PCL mat as reference), allowed us to quantify EO content within these compositions (Appendix A).

GC-MS calibration curves were prepared using increasing concentrations of the oils, tested at the same conditions as the mats (Appendix A). From the GC data, the spectrum of each oil and mat was obtained evidencing a major representative peak, with the respective characteristic RT, of citral for LGO and of 1,8-cineole for ELO and NO (Appendix A, unloaded PCL was used as control and subtracted from the PCLaEOs and PCLbEOs spectra). Data attested to the presence and release of the EOs from both PCLaEOs and PCLbEOs mats in an environment mimicking actual breathing conditions (Table 4). EOs release from PCLaNO and PCLaELO was barely detected; yet, on PCLaLGO EOs release reached 50% of the total loading mass, which may be extremely high considering the application. PCLbEOs mats behaved in a more homogeneous way, with EOs release varying between 10 and 13% after the 4 h testing. These results demonstrate that blending EOs with the polymer solution may be a more effective strategy to guarantee reproducibility and EOs controlled release over time. Authors also consider these amounts acceptable so as not to overwhelm odor-sensitive users; still, further testing is required.

### 3.5. Sensory Evaluation

To establish the EOs release amount from the absorbed and blended mats as acceptable for users, sensory evaluations, made of descriptive, discriminatory, and affective analyses, were conducted. A group of 45 people aged between 18 and 59 years (average of 30 years) and made of 60% females and 40% males were recruited. Only <7% of the participants presented respiratory/smelling issues, were smokers or had been infected with COVID-19, factors that could compromise their odor perception and sensitivity.

In a first approach, participants were requested to organize the seven samples in the study by order of odor intensity. Of these, 48.9% of participants placed PCL as the sample with the lowest odor intensity, while 44.4% selected PCLaLGO as the highest (Figure 3a), which corroborates the results from Table 4. Yet, when comparing odor intensity between groups of samples treated with the same EOs but loaded with different strategies, more than ≈75% of the participants determined the absorbed samples as the ones with the most intense smell (Figure 3b). Considering the sensory information was collected during only a few minutes of testing (<10 min) and that GC-MS spectra averaged accumulated data of a 4 h period, it is likely that EOs release from PCLaNO and PCLaELO was more important during the first instances of use and to fade away overtime. Finally, participants were requested to evaluate the mats by the pleasantness of smell. PCLbLGO was selected as the most pleasant of the group. PCLaLGO also gathered very positive feedback; however, some participants complained about the overwhelming odor and 4.4% of the those even said it was unpleasant. PCLaELO was categorized as extremely unpleasant or unpleasant by 80% of the participants.

### 3.6. Thermal Stability

Polymer-related degradation steps of unloaded and EOs-loaded PCL fibers were revealed via TGA (Figure 4) as the temperature increased.

The first degradation step on TGA spectra was identified at ≈60 °C for all samples and was associated with water molecules’ evaporation. A maximum of 5% reduction in the samples’ weight was registered, particularly for PCL and PCLaNO. Despite the characteristic volatile nature of the EOs, which were expected to evaporate together with water, results suggest that the interactions established with the polymeric fibers have increased the thermal stability of the mats protecting both polymer and oil [11,26]. EOs have been reported to induce a plasticizing effect on polymers, shifting degradation to higher rates [75]. Blending the EOs with PCL was deemed the most effective strategy for establishing stronger interactions that protected the fibers for longer. The second degradation step was initiated at ≈320 °C. Here, PCLaNO and PCLaELO were the first to experience an important mass loss, indicating that the interactions made with the PCL fibers were not as strong as with the other oils (i.e., LGO). After this point, all fibers behaved similarly as degradation targeted mostly the PCL polymeric backbone, initiating with the side chains and progressing towards the main chain scission until ≈415 °C [71]. From this temperature until 500 °C, the continuing weight loss verified can be ascribed to further degradation of the polymers into carbon char (<3%, Figure 5b).

DSC spectra were also collected to comprehend the physical and chemical transformations that the mats underwent when subjected to increasing heat (Figure 5a). DSC thermograms reported two very distinctive peaks, which corroborated the TGA analyses. The first endothermic peak was detected at ≈60 °C and was attributed to the melting point of PCL [76]. The second peak was attributed to the cleavage and degradation (Td) of the main backbone and side chains of PCL [71]. Figure 5b shows that Tm slightly increases with the introduction of the oils. This may demonstrate that strong bonds are established between the polymeric chains and the EOs, contributing to a greater packaging of the chains and, consequently, to an increase in the crystallinity of the structure [77]. This is also supported by the Tm enthalpies (ΔH), which increased with oils additions (in most cases), revealing that greater energy is required to introduce disorganization in the structure. Even though differences between samples were not substantial, the data suggest that LGO promoted the strongest bonds (increased ΔH over pristine PCL), while ELO (blending) and NO (physical adsorption) experience the lowest affinity to the PCL fibers (data also corroborated by EOs loading, Table 4). In the degradation step (Td), a decrease in the ΔH was registered for all mats containing oils. Once again, the highest enthalpy of degradation was denoted for PCLbLGO, confirming the strong affinity between this oil and the PCL fibers.

### 3.7. Mechanical Performance

The mats breaking strength and elongation at break were analyzed using rectangular-shaped specimens of 6 cm long and 2 cm width, and average thickness of ≈0.09 mm. The impact of the incorporated EOs on the mechanical performance of the mats was noticeable on both PCLaEOs and PCLbEOs (Table 5). Breaking strength and elongation at break were seen to improve as the bonds with the oils strengthened. As evidenced by the thermal stability testing, blending of EOs with PCL solutions increases mats’ protection. Here too, an improvement in mechanical resilience is noticeable from PCLaEOs to PCLbEOs. These results demonstrate that the mechanical features of the mats are governed by the EOs loading strategy. It also corroborates our earlier premise (Section 3.4) in which we categorized LGO as the EO with the highest affinity towards PCL, being capable of promoting the strongest bonds.

In the case of the PCLaEOs mats, the EOs introduced a plasticizing effect via lubrication, increasing polymer chain mobility and possibly disrupting pre-existent intermolecular bonds, which then explains a drop in the breaking strength but a significant improvement in the elasticity of the mats. Similar observations were made by Ahmed et al., when examining the effects of clove oil on polylactide/PCL films [78]. On PCLbEOs mats, bonds between polymer and oils were established in a free, unrestrained environment of a polymer blend. Here, intermolecular interactions were allowed to take place in a 24 h period, in which the biomolecules and the polymer chains were completely free and mobile to adapt to each other, without the restrains of a pre-formed, already structurally organized mat. Furthermore, it has been shown that high plasticizer concentrations (i.e., 10% VC) can lead to stronger interactions between plasticizer and polymer molecules, hindering macromolecular mobility or phase separation, thus giving rise to mats more mechanically resilient [79,80]. Finally, it is also very important to mention that smaller fiber diameters (Figure 1) have been shown to correlate with improved mechanical features [81]. Our data corroborate this statement, by demonstrating that PCLbLGO, the mat with the smallest fiber diameters (0.75 ± 0.18 µm), achieved the highest breaking strength (≈5.99 MPa) and elongation at break (≈383.70%). In face mask production, mechanical examinations are crucial to assess the engineered mat endurance to movements without tearing.

### 3.8. Wettability and Degree of Swelling

Water contact angle determinations were made on unloaded and EOs-loaded PCL mats (Table 6). Data reported an average contact angle of ≈89° for pristine PCL and above 150° for EOs-modified mats. Once again, the successful incorporation of EOs within the electrospun PCL fibers is attested. The hydrophobic nature of PCL was corroborated [37]. More importantly, the addition of the EOs altered the nature of the PCL fibers from hydrophobic to superhydrophobic (>150°) [82]. Each one of the tested EOs has in its composition important chemical elements that confer a hydrophobic profile to the molecule; for instance, terpenes, such as citral for LGO or 1,8-cineole for NO and ELO, are known for their hydrophobic character [73,83]. As the main components of the selected oils, it would be expected their addition would raise the hydrophobicity of the mats. In light of the envisaged application, this superhydrophobicity is considered a very important feature [82], as coughing or sneezing-derived droplet repellency is the main goal.

Most mats reported similar wettability between blended and physically adsorbed strategies for the same oil. Yet, an exception should be highlighted, the NO-modified mats. The hydrophobic contribution of this oil for the adsorbed mats is almost 20° lower than on the blended configuration. This is likely a result of the uneven distribution of the NO molecules on the adsorbed mat and/or the formation of weaker bonds, leading to oil displacement or removal during washings. This could also explain the observations made from the UV-visible readings of EOs loading, in which adsorption of NO was one of the lowest (Table 4), or the TGA spectra, in which the PCLaNO mats were one of the first to experience a mass loss with temperature rising (Figure 4).

Aside from the mats’ wettability, the fibers’ abilities to absorb and/or store water molecules arising from the external environment or from the water vapor generated during respiratory events are most important for successful mask production. Therefore, mats were investigated for their swelling capacity by measuring variations in weight following immersion in dH_2_O, for 4 h at RT (period defined as maximum recommend for mask use). In the end, it was seen that all samples retained their structural integrity (no reported alterations in sample size or thickness, data not shown). Pristine mats experienced the largest incorporation of water molecules (DS ≈ 42%). A result that can be easily explained by the increased wettability of this mat compared to the EOs-modified samples. Similarly, PCLaNO reported the second largest DS, again demonstrating that PCL interactions with NO were not as successful as with the other oils. The mats’ porosity is also considered a decisive factor for water binding, by restricting the space available around fibers. In this case, however, DS behaved inversely to the mats’ porosity (Figure 1). Data also support the effectiveness of the blending strategy, the superhydrophobic nature of which reduced water retention the most [84]. Considering repellence of viral particles arising from the exterior is primordial in COVID-19 protective gear, this lack of affinity towards water molecules should be considered an asset. However, knowing that viral particles are expelled by an infected user, it is important to still retain some capacity to absorb them to avoid leaks to the surrounding environment.

### 3.9. Air and Water Vapor Permeabilities

Breathability, as defined by the standard EN 14683, is related to the active degree of ventilation of specialized materials, in other words, their ability to permeate air (air flow through a fiber-based sample). According to European standards, the user’s comfort while utilizing a community face mask is guaranteed with an air permeability of at least 8 L/min. This allows an easy transfer of air from the user’s face to the external environment while preventing the penetration of water droplets [85]. Breathing creates a micro-climate between the mask and the user’s face (exhaled air and water vapor). Thus, if an ineffective material is employed in mask production, excess humidity and heat may accumulate at the interface, causing an eventual soaking of the mask, reducing filtration efficiency, and raising user discomfort.

Mats were analyzed for their air permeability by applying the recommended testing pressure of 40 Pa/cm^2^. Data reported air permeability inferior to 8 L/min (except for the mask with a ≥98% protective efficiency). Yet, when compared to commercial masks with a 95% protection efficacy (≈4.31 L/min), the EOs-loaded mats were not so ineffective; both PCLbLGO and PCLbNO reported permeabilities above 5 L/min. All EOs-loaded mats would be expected to display superior air permeability than PCL mats due to their increased porosity (Figure 1). However, enhanced porosities do not necessarily mean larger pore sizes and, hence, increased air flux. In fact, considering the intricated network of thin fibers that characterized the mats loaded with EOs, smaller pores can occur, conditioning air permeability [86]. The affinity of the EOs towards PCL (in both EOs amount and bonding strength) may also have played an important role here since the tendency was for blended mats to display higher air permeability than those modified by physical adsorption. The exception was PCLbELO, which, as seen by its loading (Table 4) and its thermal behavior (Figure 4), was the least effective in developing strong bonds with PCL.

Regarding the water vapor permeabilities, data reported an improvement in the WVTR and the permeability index (I) with the incorporation of the EOs. According to Table 7, it would be expected the presence of EOs would reduce the WVTR of the mats due to their superhydrophobicity and low degree of swelling. The addition of oils to electrospun fibers has been shown to prevent water loss when oils are located in spaces between fibers. Yet, if the oil remains on/in the fibers, WVTR is not affected by its presence [87]. Here, we hypothesize the mats’ porosity as the major contributor to the observed water vapor permeabilities. Porosity increased as follows, PCL < PCLaNO < PCLaLGO < PCLaELO < PCLbNO < PCLbLGO < PCLbELO (Figure 1), which is basically the same tendency observed for WVTR (Table 7). This increasing trend can be explained by the Fickian diffusion model, in which diffusion flux exhibits a positive linear correlation with the materials’ porosity [88]. Once again, EOs-blended mats were more permeable, and between those PCLbLGO was the most effective in facilitating water vapor flux. In fact, PCLbLGO mats promoted a permeability index ≈20% superior to the commercial surgical masks tested, which could reduce humidity accumulation and consequent breathing difficulties and discomfort frequently associated with such protective gear [89].

### 3.10. Bacteriophage Permeability and Inactivation

The bacteriophage MS2 is a non-pathogenic (to eukaryote), icosahedral capsid, positive-sense single-stranded RNA virus and, thus, a potential surrogate of SARS-CoV-2 [42,43]. Such parallelism allows testing of unloaded and EOs-loaded PCL mats against the viral particles and extrapolation of data to a real-life scenario. Mats were examined for their ability in preventing viral droplets to permeate and infect agar plates and for their capacity to kill the MS2 virus upon direct contact (Figure 6). Permeability examinations were conducted from 4 h to 24 h of incubation at 37 °C. After 4 h of culture, the 5 µL droplets were still present on top of the EOs-loaded surfaces (Appendix A). Yet, on PCL the droplets had disappeared (data not shown). Thus, when reaching 24 h of culture, it was no surprise to find viral plaques surrounding the PCL disk (Figure 6). Despite its hydrophobicity, PCL was incapable of retaining the viral load, permeating to the agar below and killing the host cells. Most importantly, all mats modified with EOs, regardless of incorporation strategy, were found successful in preventing viral penetration and subsequent infection spreading. Here, the superhydrophobic nature of the mats (Table 6), achieved through specialized EOs components, namely terpenes (i.e., citral and 1,8-cineole) [73,83], guaranteed viral repellency.

Considering SARS-CoV-2 virions are ≈50–125 nm in size and are frequently transmitted in the form of aerosolized small droplets (<10 µm) [90], it is conceivable that some of the viral load deposited on top of the mats to infiltrate the surface via pores. Furthermore, fiber swelling via water adsorption is unavoidable (Table 6). Hence, it was vital to understand the EOs antimicrobial effect against the bacteriophage MS2 while this intimate contact was maintained. Contact inactivation experiments were conducted for 4 h, using mats incorporated with 10% VC of each oil and incubated with a 50 µL virus-charged droplet (Appendix A). As expected, PCL had the smallest impact on the viral load. The ≈0.27 log reduction observed can be explained by the retention of the viral particles by the mats’ fiber entangled morphology. Mats modified via blending were more effective in inactivating the MS2 virus. Using this strategy is likely that the distribution of the antimicrobial agents covered a larger portion of the mat (more abundant), increasing fiber homogeneity and raising virucidal performance. This could also explain the smaller impact of the PCLaNO mat since interactions between NO and PCL were not the most successful (as verified by TGA data). Interestingly, despite the significant reduction in the EOs concentration required to achieve a complete inactivation of the viral load, PCLaEOs and PCLbEOs mats were still able to reach ≈0.85 and ≈1.07 log reduction, respectively.

It should be stressed that experiments were conducted in extreme scenario conditions (large volume droplets, very unlikely to occur during coughing or sneezing-like events), so an informed selection of the most effective combination could be made. From the viral repellency and killing testing, the most effective modification and incorporated biomolecule was the blending of PCL with LGO (PCLbLGO—absence of viral plate formation and ≈1.36 log reduction, which can be translated to >90% inactivation). It is possible that the LGO mechanisms of action could have an important contribution here since, contrary to NO and ELO which effect is the most important upon host penetration [57], LGO acts by surrounding the viral particles prior to any interference with the host [54]. Considering viral droplets cannot easily penetrate the mats barrier because of their superhydrophobicity, being retained at the surface, during that contact period LGO can generate an antimicrobial coating around the virus, hence, inhibiting specific binding with the host and, consequent, infection spreading. Through this, LGO can initiate its antimicrobial effects earlier than both NO and ELO.

## 4. Conclusions

The COVID-19 pandemic has forced the adoption of new protection habits that include face mask-wearing. This research has demonstrated the potential of EOs loaded onto PCL electrospun mats to work as COVID-19 active barriers and offer the population another antiviral protective option.

LGO, NO and ELO were highlighted from a group of 20 antimicrobial EOs for their inhibitory effect against the *E. coli* MS2 virus, a surrogate of COVID-19. PCL was successfully modified with the oils via absorption and blending, exhibiting a superhydrophobic nature, capable of preventing droplet penetration, and improved mechanical and thermal resilience. Blending allowed an even distribution of the oils, which guaranteed stronger interactions between oils and polymer to be formed (i.e., PCLbLGO), smaller fiber diameters and consequently larger porosities, which improved air and water-vapor permeabilities. Antiviral evaluations demonstrated the EOs-loaded mats’ ability to inhibit viral particle infiltration and activity. PCLbLGO was deemed the most effective mat from the group, guaranteeing a ≈1.36 log reduction of MS2 activity. This mat was also regarded as the most odor-pleasant during sensory evaluation and, as such, a potentially simple and low-cost option to be taken into consideration in future face mask production.

## Figures and Tables

**Figure 1 pharmaceutics-14-00303-f001:**
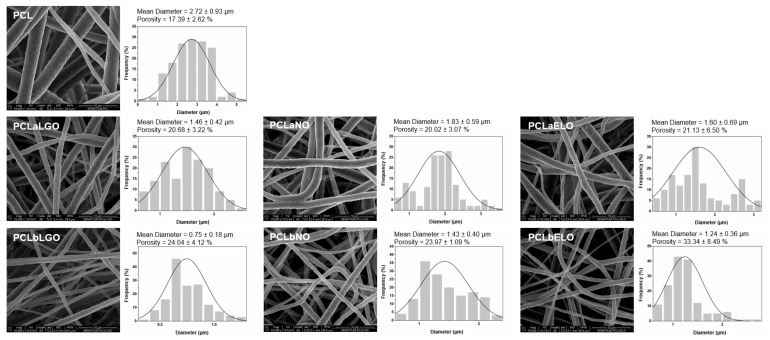
FEG-SEM micrographs and diameter distribution histograms of the PCL-based electrospun mats (unloaded, PCL; loaded with EOs by physical adsorption, PCLaLGO, PCLaNO and PCLaELO; and loaded with EOs by blending, PCLbLGO, PCLbNO and PCLbELO), with identification of average diameters and porosity. Scale bars in the micrographs represent 10 µm.

**Figure 2 pharmaceutics-14-00303-f002:**
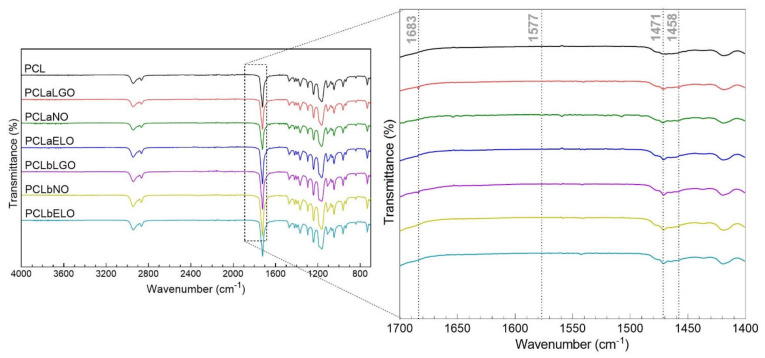
ATR-FTIR spectra of the PCL, PCLaEOs and PCLbEOs mats collected between 4000–700 cm^−1^. As the most significant for this study, the section between 1700–1400 cm^−1^ was amplified.

**Figure 3 pharmaceutics-14-00303-f003:**
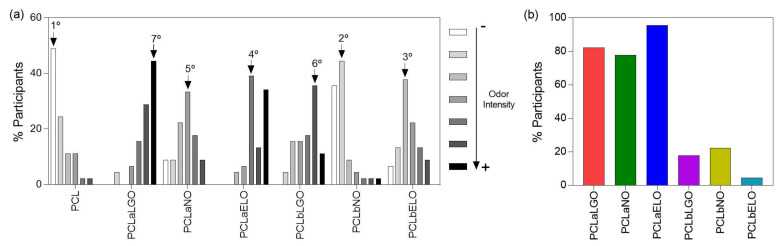
(**a**) Classification of the mats based on their odor intensity level. Participants were required to select one mat for each level of intensity (100% participation per level). Highest classifications for each mat are identified with an arrow and ranking position. (**b**) Discriminatory evaluation between mats modified with the same oil via different loading strategies (PCLaEOs vs PCLbEOs).

**Figure 4 pharmaceutics-14-00303-f004:**
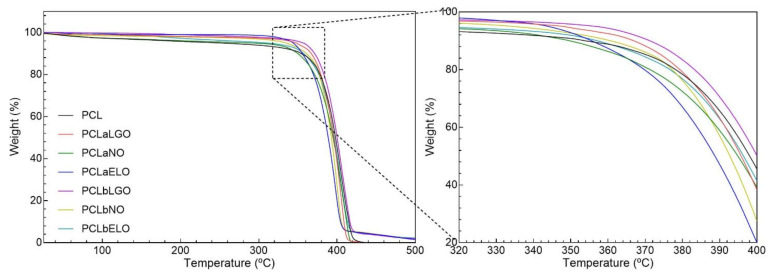
TGA spectra of the PCL-based mats obtained from 25 to 500 °C under nitrogen atmosphere, flow rate of 200 mL/min and temperature rise of 10 °C/min.

**Figure 5 pharmaceutics-14-00303-f005:**
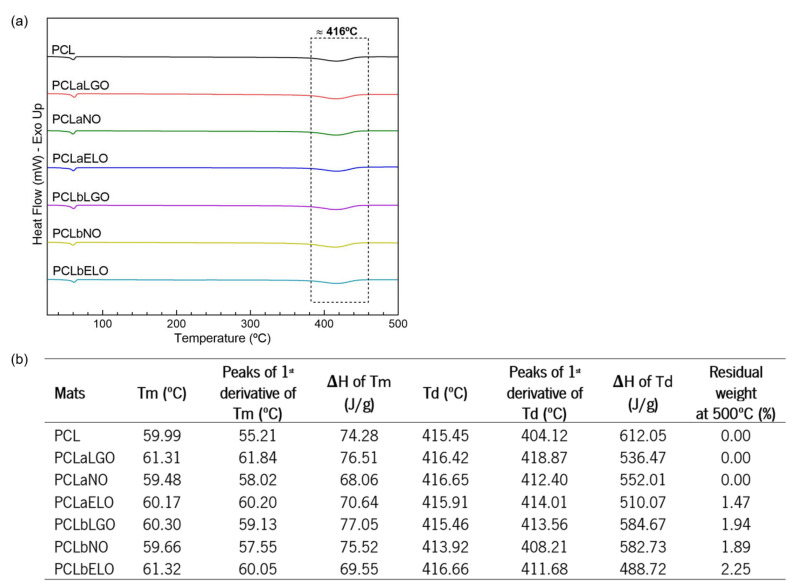
(**a**) DSC spectra of the PCL-based mats obtained from 25 to 500 °C under nitrogen atmosphere, flow rate of 200 mL/min and temperature rise of 10 °C/min. (**b**) Main DSC thermal transitions (*Tm*, melting temperature, *Td*, degradation temperature, and respective *Δ**H*, enthalpy change) and TGA data (1st derivative temperature peaks, and residual weight at 500 °C), with *n* = 3 and S.D. < 1%.

**Figure 6 pharmaceutics-14-00303-f006:**
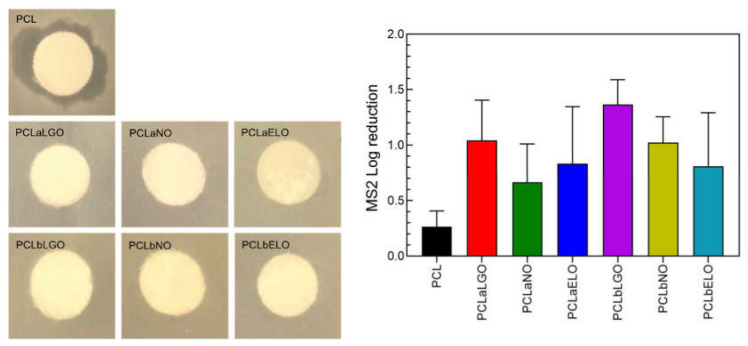
Inhibition of bacteriophage MS2 activity by unloaded and EOs-loaded PCL fibers examined via (**left**) mat permeability (24 h) and (**right**) direct contact inactivation (4 h). EOs diffusion from the mats to the agar plates was not observed during permeability testing. Interference of EOs on the host during contact inactivation experimentation was taken into consideration and accounted for in the present data. LGO and NO introduced a ≈2% and a ≈5% host-related influence, respectively, while ELO did not affect the host.

**Table 1 pharmaceutics-14-00303-t001:** List of tested EOs (all extracted via azeotropic distillation processes), their origin and density, as well as maximum (50% *v*/*v*) and minimum (1.25% *v*/*v*) tested concentrations.

EO	Origin (Family)	Density (g/cm^3^)	Maximum Concentration (mg/mL)	Minimum Concentration (mg/mL)
Amyris (AO)	*Amyris balsamifera* Linn. (Rutaceae)	0.957	78.50	11.96
Cajeput (CJO)	*Melaleuca leucadendron* Linn. (Myrtaceae)	0.911	455.50	11.39
Cinnamon leaf (CLO)	*Cinnamomum zeylanicum* Blume (Lauraceae)	1.049	524.50	13.11
Citronella (CIO)	*Cymbopogon winterianus* Jowitt (Poaceae)	0.882	441.00	11.03
Clove (CO)	*Eugenia caryophyllus* (Spreng.) Bullock & S.G.Harrison (Myrtaceae)	1.056	528.00	13.20
Eucalyptus (ELO)	*Eucalyptus globulus* Labill. (Myrtaceae)	1.465	732.50	18.31
Frankincense (FO)	*Boswellia carterii* Birdw. (Burseraceae)	0.857	428.50	10.71
Geranium (GO)	*Pelargonium graveolens* (Thunb.) L’Hér. (Geraniaceae)	0.895	447.50	11.19
Himalayan cedar (HCO)	*Cedrus deodara* Loud. (Pinaceae)	0.935	467.50	11.69
Lavandin (LO)	*Lavandula hybrida* Balb. (Lamiaceae)	0.889	444.50	11.11
Lemongrass (LGO)	*Cymbopogon flexuosus* (Nees ex Steud.) W. Watson (Poaceae)	0.890	445.00	11.13
Niaouli (NO)	*Melaleuca viridiflora* Sol. ex Gaertn (Myrtaceae)	0.913	456.50	11.41
Orchid (OO)	*Serapias vomeracea* (Burm.f.) Briq. (Orchidaceae)	0.856	428.00	10.70
Palmarosa (PMO)	*Cymbopogon martinii* (Roxb.) Wats. var. motia Burk. (Poaceae)	0.884	442.00	11.05
Patchouli (PTO)	*Pogostemon patchouli* Hook.fil. (Lamiaceae)	0.960	480.00	12.00
Rosemary (RO)	*Rosmarinus officinalis* Linn. (Lamiaceae)	0.900	450.05	11.25
Sage (SO)	*Salvia officinalis* Linn. (Lamiaceae)	0.915	457.50	11.44
Star anise (SAO)	*Illicium verum* Hook.fil. (Schisandraceae)	0.981	490.50	12.26
Tea tree oil (TTO)	*Melaleuca alternifolia* (Maiden & Betche) Cheel (Myrtaceae)	0.895	447.50	11.19
Wintergreen (WO)	*Gaultheria procumbens* Linn. (Ericaceae)	1.182	591.00	14.76

**Table 2 pharmaceutics-14-00303-t002:** MBCs and VCs of selected EOs against the bacterium *E. coli* (host) and the bacteriophage MS2 (*n* = 3, S.D. < ± 5.0 mg/mL). Listing of EOs was performed in descent order with respect to the acquired VC.

EOs	MBC (mg/mL)	VC (mg/mL)
LGO	178.0	356.0
NO	45.7	365.2
OO	85.6	428.0
TTO	22.4	447.5
CO	105.6	528.0
ELO	>732.5	586.0

**Table 3 pharmaceutics-14-00303-t003:** Determination of conductivity and viscosity parameters associated with the PCL-based solutions prepared in 9/1 *v*/*v* CHF/DMF ratio and used for electrospinning (*n* = 3).

Polymeric Blends	Conductivity (µS/cm)	Viscosity (Pa.s)
PCL	0.225 ± 0.011	1.164 ± 27
PCLbLGO	0.093 ± 0.005	1.105 ± 26
PCLbNO	0.078 ± 0.016	1.676 ± 86
PCLbELO	0.150 ± 0.009	0.981 ± 22

**Table 4 pharmaceutics-14-00303-t004:** EOs mass (µg) detected on film samples of 6 mm in diameter via UV-visible spectroscopy. Mass determinations for PCLaEOs were made indirectly, by measuring the absorbances of the EOs loading solution after 24 h mat immersion and of the ethanol washing bath (desorption of loosely bound EOs molecules). EOs release was mapped by GC-MS and normalized to the most important compound within the EOs, namely citral and 1,8-cineole. Data are reported as mean ± S.D. (*n* = 3).

Mats	Loaded Mass (µg)	EOs Release (%)
PCLaLGO	5257 ± 2510	50.3 ± 0.1
PCLaNO	800 ± 184	0.3 ± 0.1
PCLaELO	5079 ± 4122	0.1 ± 0.0
PCLbLGO	1673 ± 641	13.3 ± 3.5
PCLbNO	1128 ± 98	11.3 ± 8.4
PCLbELO	631 ± 173	10.7 ± 6.3

**Table 5 pharmaceutics-14-00303-t005:** Unloaded and EOs-loaded PCL mats mechanical examinations. Data are reported as mean ± S.D. (*n* = 3).

Mats	Breaking Strength (MPa)	Elongation at Break (%)
PCL	2.08 ± 0.85	207.33 ± 64.26
PCLaLGO	1.44 ± 0.36	271.90 ± 53.03
PCLaNO	0.92 ± 0.04	244.00 ± 118.65
PCLaELO	1.89 ± 0.40	294.35 ± 137.11
PCLbLGO	5.99 ± 0.82	383.70 ± 40.10
PCLbNO	2.95 ± 0.74	358.40 ± 128.50
PCLbELO	3.52 ± 0.81	330.33 ± 109.13

**Table 6 pharmaceutics-14-00303-t006:** Contact angles and degree of swelling determinations using dH_2_O as testing solution. Data is reported as mean ± S.D. (*n* = 6).

Mats	Contact Angle (°)	Degree of Swelling (%)
PCL	88.95 ± 4.17	42.34 ± 5.41
PCLaLGO	160.03 ± 4.21	22.42 ± 3.43
PCLaNO	156.80 ± 4.10	32.48 ± 8.16
PCLaELO	156.18 ± 3.27	26.32 ± 7.29
PCLbLGO	159.05 ± 7.24	11.86 ± 3.52
PCLbNO	176.97 ± 4.03	23.10 ± 2.29
PCLbELO	163.95 ± 5.68	17.67 ± 3.21

**Table 7 pharmaceutics-14-00303-t007:** Air and water vapor permeabilities, with determination of *WVTR* (g/m^2^.day) and *I* (%) for the last.

Mats	Air Permeability (L/min)	*WVTR* (g/m^2^.day)	*I* (%)
PCL	4.33 ± 0.65	21,314.03 ± 6428.26	95.50 ± 20.33
PCLaLGO	3.79 ± 0.68	26,069.58 ± 10,668.75	116.80 ± 33.75
PCLaNO	2.71 ± 0.45	24,598.99 ± 27.01	110.23 ± 0.13
PCLaELO	3.29 ± 0.52	26,289.21 ± 3470.72	117.80 ± 10.95
PCLbLGO	5.85 ± 1.14	28,991.66 ± 8048.83	129.90 ± 25.45
PCLbNO	5.15 ± 0.56	26,518.40 ± 10,088.04	118.81 ± 31.91
PCLbELO	3.43 ± 0.88	27,186.85 ± 7522.14	121.81 ± 23.78
≥95% Surgical Mask	4.31 ± 0.50	24,554.42 ± 473.37	109.98 ± 2.12
≥98% Surgical Mask	8.00 ± 0.25	24,096.06 ± 3071.63	107.93 ± 13.76
Standard Testing Fabric	-	22,319.89 ± 11.03	100.00 ± 0.00

## Data Availability

Not applicable.

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
