# Peer review of "Inhibition of Escherichia Virus MS2, Surrogate of SARS-CoV-2, via Essential Oils-Loaded Electrospun Fibrous Mats: Increasing the Multifunctionality of Antivirus Protection Masks"

_pharmaceutics, 2022, doi:10.3390/pharmaceutics14020303_

Round 1

Reviewer 1 Report

Dear Authors,

Your manuscript is interesting and up-to-date. The COVID-19 pandemic has forced the adoption of new protection habits that include face mask wearing. Face mask usage is one of the most important measures employed to reduce SARS-CoV2 transmission at the moment. The authors offer a very original solution including antiviral essential oils (EOs) within polycaprolactone (PCL) electrospun fibrous mats to be used as intermediate layers in individual protection masks. From the analyzes of 20 essential oils, the authors found that only three of them have the appropriate antiviral properties.
In my opinion, the introduction is completely accurate and clear. Most of the sources cited by the authors are 3-5 years old, there is even one of the recent new 2022. Citing many new literature sources shows how seriously and wholeheartedly the authors have taken the research they have done.
In the Methods and Material section, the authors have described in great detail and clearly the methods they have used and any researcher could replicate them.
In the section Results and discussion the results are presented visually and are supported by the respective tables and figures. The results obtained from each analysis are clear and accurate and are supported by serious research discussion and citations.
The conclusion supports the results obtained by the authors.
I would like to ask the authors to review their manuscript very carefully for typographical errors and omissions. In the 76 source cited by the authors, they missed the year and the pages.

Author Response

Reviewer #1

Your manuscript is interesting and up-to-date. The COVID-19 pandemic has forced the adoption of new protection habits that include face mask wearing. Face mask usage is one of the most important measures employed to reduce SARS-CoV2 transmission at the moment. The authors offer a very original solution including antiviral essential oils (EOs) within polycaprolactone (PCL) electrospun fibrous mats to be used as intermediate layers in individual protection masks. From the analyzes of 20 essential oils, the authors found that only three of them have the appropriate antiviral properties.

In my opinion, the introduction is completely accurate and clear. Most of the sources cited by the authors are 3-5 years old, there is even one of the recent new 2022. Citing many new literature sources shows how seriously and wholeheartedly the authors have taken the research they have done. In the Methods and Material section, the authors have described in great detail and clearly the methods they have used and any researcher could replicate them.

In the section Results and discussion the results are presented visually and are supported by the respective tables and figures. The results obtained from each analysis are clear and accurate and are supported by serious research discussion and citations. The conclusion supports the results obtained by the authors.

I would like to ask the authors to review their manuscript very carefully for typographical errors and omissions. In the 76 source cited by the authors, they missed the year and the pages.

R. We appreciate the Reviewers insights into our manuscript and the criticism made. We have made a thorough revision of the manuscript, tracking English writing/grammar mistakes and correcting them. All references have been updated and revised.

Reviewer 2 Report

To the Authors (in detail):

  • Introduction section, the Authors have to explain why they have used the 20 species listed in table 1 for this experiment, in addition they have to include, if possible, some reference about the effect anti-bacteria or anti-viral previously studied for each species;
  • Introduction section. This section has to be improved, you have to explain that the essential oil composition of a part of a plant is influenced by many factors such as harvest year [X1], harvest date [X2], cultivation condition [X3], geographical area of cultivation [X4], variety or cultivar if it is a cultivated plant [X5], age of plant or parts of plant [X6], essential oil extraction system [X7], …..and so on. Support this statement, at least, with the following references, … if possible one for each factor influencing the essential oil composition.

[X1] The peel essential oil composition of bergamot fruit (Citrus bergamia, Risso) of Reggio Calabria  (Italy): a review.

Emirates Journal of Food and Agriculture  32 (11) 835-845 (2020)

[X2] Variations of Essential Oil Constituents in Oregano (Origanum vulgare subsp. viridulum (= O. heracleoticum) over Cultivation Cycles.

Plants 2020, 9, 1174; doi:10.3390/plants9091174

 [X3] Chemical components of essential oils and biological activities of the aqueous extract of Anethum graveolens L. grown under inorganic and organic conditions.

Chem. Biol. Technol. Agric. (2021) 8:20 https://doi.org/10.1186/s40538-021-00224-9

[X4] Variability in essential oil composition, antioxidant and antimicrobial activities of Ruta montana L. collected from different geographical regions in Algeria.

Journal of Essential Oil Research Volume 32, 2020 - Issue 1. https://doi.org/10.1080/10412905.2019.1660238

[X5] Citrus bergamia, Risso: the peel, the juice and the seed oil of the bergamot fruit of Reggio Calabria (South Italy).

Emirates Journal of Food and Agriculture 32(7) 522-532 (2020).

DOI: 10.9755/ejfa.2020.v32.i7.2128

[X6] Plant Shoot Age and Temperature Effects on Essential Oil Yield and Oil Composition of Rose-Scented Geranium (Pelargonium sp.) Grown in South Africa.

Journal of Essential Oil Research Volume 18, 2006 - Issue sup1.

 https://doi.org/10.1080/10412905.2006.12067129

[X7] A study on the extraction of essential oil of Persian black cumin using static supercritical CO2 extraction, and comparison with hydro-distillation extraction method

Separation Science and DOI

10.1080/01496395.2018.1541907Technology (Philadelphia)Volume 54, Issue 11, Pages 1778 - 178624 July 2019

  • 1 section and in the whole manuscript, tables and figures, when you indicate the temperature, separate the numeric value from the symbol: 121 °C and not 121°C;
  • 1 section, please, can you include one column in the table 1, to indicate the extraction method?
  • 1 section, can you indicate the production year and the geographical origin of each essential oil?
  • The scientific names are sometime incorrectly written, please, verify on: www.gbif.org;
  • Table 1 do not contain detailed information; the botanical Origin column has to be revised. For each species has to be included the botanist and the family (the family in brackets);
  • Table 1, Cedros deodora Loud, the name of the botanist has not to be italicized;
  • Table 1, for Orchidaceae you have written only the family, but the genus and the species are missed. In addition, you have to use the binomial botanical International system, the family has not to be italicized;
  • Table 1, Palmarosa line, what is: Cymbopogon 4nactiv var.motia. Please verify carefully and write using the botanical International system:
  • In your discussion section you have mainly discussed about 1,8-cineole. May you discuss also the effect of some other molecule? (include references);
  • References section, ref 62, only the first letter of the name of each author has to be written in capital letter;
  • References section, refs 23, 24, 25, 48, 57, 66, 67, 72 and in the whole manuscript: the scientific name has to be italicized:
  • References section, ref 76, verify the instructions for authors and complete data to find this reference;
  • References section, the number of the last page of this paper is missed;
  • References section. This section has to be arranged in light of the instructions for authors of Pharmaceutics. For example, the journal name has to be abbreviated. Please, see: https://www.library.caltech.edu/journal-title-abbreviations;
  • References section: sometime you have written the title of the paper with the first letter of each word in capital letter and sometime non. Please be consistent in the whole section and use some recently published paper as a template;
  • References section, ref 83: for the journal name, the first letter of each word in capital letter;
  • References section, ref 42, Complete the necessary information;
  • Please, very important, write in blue color or evidence differently the corrections you will do.

Author Response

Reviewer #2

To the Authors (in detail):

Introduction section, the Authors have to explain why they have used the 20 species listed in table 1 for this experiment, in addition they have to include, if possible, some reference about the effect anti-bacteria or anti-viral previously studied for each species.

R. The Reviewer is completely right. We apologize for this lack of information. Instead of adding that information in the Introduction, we added it in the Materials and Methods section, sub-section 2.1. We used one publication of our group to justify their selection, which was done based on their previously identified antibacterial potential.

Introduction section. This section has to be improved, you have to explain that the essential oil composition of a part of a plant is influenced by many factors such as harvest year [X1], harvest date [X2], cultivation condition [X3], geographical area of cultivation [X4], variety or cultivar if it is a cultivated plant [X5], age of plant or parts of plant [X6], essential oil extraction system [X7], …..and so on. Support this statement, at least, with the following references, … if possible one for each factor influencing the essential oil composition.

[X1] The peel essential oil composition of bergamot fruit (Citrus bergamia, Risso) of Reggio Calabria  (Italy): a review.

Emirates Journal of Food and Agriculture  32 (11) 835-845 (2020)

[X2] Variations of Essential Oil Constituents in Oregano (Origanum vulgare subsp. viridulum (= O. heracleoticum) over Cultivation Cycles.

Plants 2020, 9, 1174; doi:10.3390/plants9091174

[X3] Chemical components of essential oils and biological activities of the aqueous extract of Anethum graveolens L. grown under inorganic and organic conditions.

Chem. Biol. Technol. Agric. (2021) 8:20 https://doi.org/10.1186/s40538-021-00224-9

[X4] Variability in essential oil composition, antioxidant and antimicrobial activities of Ruta montana L. collected from different geographical regions in Algeria.

Journal of Essential Oil Research Volume 32, 2020 - Issue 1. https://doi.org/10.1080/10412905.2019.1660238

[X5] Citrus bergamia, Risso: the peel, the juice and the seed oil of the bergamot fruit of Reggio Calabria (South Italy).

Emirates Journal of Food and Agriculture 32(7) 522-532 (2020).

DOI: 10.9755/ejfa.2020.v32.i7.2128

[X6] Plant Shoot Age and Temperature Effects on Essential Oil Yield and Oil Composition of Rose-Scented Geranium (Pelargonium sp.) Grown in South Africa.

Journal of Essential Oil Research Volume 18, 2006 - Issue sup1.

https://doi.org/10.1080/10412905.2006.12067129

[X7] A study on the extraction of essential oil of Persian black cumin using static supercritical CO2 extraction, and comparison with hydro-distillation extraction method

Separation Science and DOI 10.1080/01496395.2018.1541907Technology (Philadelphia)Volume 54, Issue 11, Pages 1778 - 178624 July 2019

R. We appreciate the detailed information provided by the Reviewer, who is clearly an expert in this field. As such, we have followed the Reviewer expert guidelines and added the details listed above, using the references provided to support them.

1 section and in the whole manuscript, tables and figures, when you indicate the temperature, separate the numeric value from the symbol: 121 °C and not 121°C;

R. As recommended, an extra space between the sign and the number has been added.

1 section, please, can you include one column in the table 1, to indicate the extraction method?

1 section, can you indicate the production year and the geographical origin of each essential oil?

R. We were able to contact the supplier about these issues, however they were only keen to provide the information concerning the extraction method but not about the year or geographical origin of each EOs. We apologize for that. We have added that information to the subtitle of Table 1 since the information we received is that the same process was used for all EOs.

The scientific names are sometime incorrectly written, please, verify on: www.gbif.org;

Table 1 do not contain detailed information; the botanical Origin column has to be revised. For each species has to be included the botanist and the family (the family in brackets);

Table 1, Cedros deodora Loud, the name of the botanist has not to be italicized;

Table 1, for Orchidaceae you have written only the family, but the genus and the species are missed. In addition, you have to use the binomial botanical International system, the family has not to be italicized;

Table 1, Palmarosa line, what is: Cymbopogon 4nactiv var.motia. Please verify carefully and write using the botanical International system.

R. Thank you so much for pointing this out. We have made the necessary adjustments and hope the names are now corrected. Additional information was not possible to obtain since the suppliers were not capable of providing additional data about the oils. Again, we apologize for not being able to respond to the Reviewer’s requests in their entirety.

In your discussion section you have mainly discussed about 1,8-cineole. May you discuss also the effect of some other molecule? (include references).

R. This was a deliberate choice maid by the authors. Considering this is the main component of two of our oils and the amounts of the other components are not significant or have very little impact in antiviral action, the authors have decided to focus their attention on the EOs main components. This allowed for the discussion to become more concise, direct and less tiresome for the readers. A more inclusive analysis of the different components of the oils would require as well a more in depth evaluation of their composition, something that in our view would require more testing and would alter the envisaged direction of the manuscript subject.

References section, ref 62, only the first letter of the name of each author has to be written in capital letter;

References section, refs 23, 24, 25, 48, 57, 66, 67, 72 and in the whole manuscript: the scientific name has to be italicized:

References section, ref 76, verify the instructions for authors and complete data to find this reference;

References section, the number of the last page of this paper is missed;

References section. This section has to be arranged in light of the instructions for authors of Pharmaceutics. For example, the journal name has to be abbreviated. Please, see: https://www.library.caltech.edu/journal-title-abbreviations;

References section: sometime you have written the title of the paper with the first letter of each word in capital letter and sometime non. Please be consistent in the whole section and use some recently published paper as a template;

References section, ref 83: for the journal name, the first letter of each word in capital letter;

References section, ref 42, Complete the necessary information;

Please, very important, write in blue color or evidence differently the corrections you will do.

R. All references have been updated following the Reviewers’ instructions and journal guidelines, and details have been added to many references.

Round 2

Reviewer 2 Report

To the Authors (in detail):

  • Please, re-edit the figures. Figure 1 (x-y axis and captions of histograms are not readable); Figure 2 (the captions inside are not readable); Figures 3-4-5 (captions inside the figure and figure 5b are not readable). The color of both words and figures is very faint;
  • Table 1, it is enough the abbreviation of the botanist, for example: Linn. and not Linnaeus;
  • References section, this section has to be cerefully revised: for example, the journals names have to be abbreviated whereas they are not;
  • Please, very important, write in red color or evidence differently the corrections you have done in your revision-1 and the corrections you will do (including the references section), to help the reviewer to find quickly your corrections.

Author Response

Reviewer #2

Please, re-edit the figures. Figure 1 (x-y axis and captions of histograms are not readable); Figure 2 (the captions inside are not readable); Figures 3-4-5 (captions inside the figure and figure 5b are not readable). The color of both words and figures is very faint;

The images have been altered as recommended. The size of the information has been increased. However, we were not able to alter the color intensity. In the software Graphpad Prism the colors are already very bright and strong. The software then adjusts the coloration of the lines as the size of the figure (regardless of type, .tif, .jpeg, etc.) increases or reduces. If the Reviewer will be so kind as too make zoom on his/her images, he/she will see that the quality of the image is significant and the data is very readable.

Table 1, it is enough the abbreviation of the botanist, for example: Linn. and not Linnaeus;

Abbreviations have now been introduced on Table 1.

References section, this section has to be cerefully revised: for example, the journals names have to be abbreviated whereas they are not;

The journal names have all been altered and abbreviated following the ISO rules.

Please, very important, write in red color or evidence differently the corrections you have done in your revision-1 and the corrections you will do (including the references section), to help the reviewer to find quickly your corrections.

We have highlighted the alterations made. However, in what respects the References, the authors have no control over the track changes. By using the software EndNote to include references and update their information, the modifications introduced become automatic and as part of the manuscript, so then we cannot highlight the alterations. We hope you can understand this.

Round 3

Reviewer 2 Report

Manuscript Number: pharmaceutics-1544233, titled:

Inhibition of Escherichia virus MS2, surrogate of SARS-CoV-2, via essential oils-loaded electrospun fibrous mats: increasing the multifunctionality of antivirus protection masks

Review 3 – 21 January 2022

Dear Editor of Pharmaceutics

the argument is interesting and the authors have improved their manuscript. I suggest the publication of this manuscript as is.

Regards.
